# Peptidyl Transferase Center and the Emergence of the Translation System

**DOI:** 10.3390/life7020021

**Published:** 2017-04-25

**Authors:** Savio Torres de Farias, Thais Gaudêncio Rêgo, Marco V. José

**Affiliations:** 1Laboratório de Genética Evolutiva Paulo Leminsk, Departamento de Biologia Molecular, Universidade Federal da Paraíba, João Pessoa 58051-900, Brazil; 2Departamento de Informática, Universidade Federal da Paraíba, João Pessoa 58051-900, Brazil; gaudenciothais@gmail.com; 3Theoretical Biology Group, Instituto de Investigaciones Biomédicas, Universidad Nacional Autónoma de México, Ciudad Universitaria, 04510 CDMX, Mexico

**Keywords:** PTC, evolution, translation, origins

## Abstract

In this work, the three-dimensional (3D) structure of the ancestral Peptidyl Transferase Center (PTC) built by concatamers of ancestral sequences of tRNAs was reconstructed, and its possible interactions with tRNAs molecules were analyzed. The 3D structure of the ancestral PTC was also compared with the current PTC of T. thermophilus. Docking experiments between the ancestral PTC and tRNAs suggest that in the origin of the translation system, the PTC functioned as an adhesion center for tRNA molecules. The approximation of tRNAs charged with amino acids to the PTC permitted peptide synthesis without the need of a genetic code.

## 1. Introduction

The origin of the translation system is a major evolutionary transition because it enabled the establishment of the ribonucleoprotein world. The transfer RNA (tRNA) molecule played a central role during the origin of translation, bridging RNA and (RNA + proteins) worlds. These tRNAs may offer clues towards the elucidation of the origin of the genetic code. The ribosome is also at the core of this process: more specifically, the Peptidyl Transferase Center (PTC) responsible for peptide bond formation. The PTC is perhaps the oldest ribozyme due to its essential function of protein synthesis and because it is highly conserved in all cellular organisms [1,2,3,4,5,6,7,8]. Since natural selection works without anticipation, the early PTC could have acted as a ribozyme that randomly produced peptide chains [9,10]. Agmon et al. [11], Agmon [12], and Bashan et al. [13] observed that the PTC possess a mirror symmetry and suggested that it must have evolved by gene fusion from two domains with similar structures, each acting as part of the catalytic center. Tamura examined the topology of the secondary structure of the PTC and suggested that it originated by gene fusions of primitive tRNA molecules [14]. The relationships between ribosomal RNAs and tRNAs had already been suggested by Bloch et al.: they analyzed the sequences of 16S ribosomal molecules and common elements were identified with tRNA molecules [15,16]. Davidovich et al. analyzed the possibility that small RNAs acquired a similar structure to that of the PTC [4]. They found that dimers of small RNAs, with stem-elbow-stem structures, showed similar structural organization with the PTC. Farias et al. reconstructed the ancestral sequences of tRNAs, and when they compared concatamers of these tRNAs with the sequence of PTC from *Thermus thermophilus*, a similarity of 50.52% was found [8]. Therefore, they suggested that the PTC arose from the junction of proto-tRNAs. Root-Bernstein and Root-Bernstein conducted analyses of sequences and secondary structures of the ribosome and suggested that it was derived from a collection of tRNAs which could have functioned as a primitive genome [17]. These data suggest that there is a common structural identity between tRNAs and the PTC. Several models about the origins and evolution of ribosomes have been proposed [4,8,9,10], yet no consensus has been reached. Thus, the search for an explanation for one of the most important evolutionary transitions in the origins of life, the emergence of ribosome, remains elusive.

In this work, the three-dimensional structure (3D structure) of the ancestral PTC built by concatamers of ancestral sequences of tRNAs was reconstructed, as proposed by Farias et al., to test its possible interactions with tRNAs anticodon loop. The 3D structure of the ancestral PTC with the current PTC of *T. thermophiles* [8] was also compared.

This study discusses the present results in terms of the possible functional relationships in the primitive translation system.

## 2. Materials and Methods

The reconstruction of the ancestral 3D structure of PTC was carried out by using the sequence proposed by Farias et al. [8]. The 3D structure used as a template control was the ribosome from *T. thermophilus* obtained from the Protein Data Base (ID4V8X) [18]. The ancestral sequence of PTC and the current *T. thermophilus* ribosome sequence were aligned and the 3D structure was modeled using the ModeRNA Server [19]. After reconstruction of the 3D model of the ancestral PTC, the structure of the model was aligned with the template of PTC from *T. thermophilus* (ID4V8X) using the Setter program [20]. For the docking experiments, the 3D structures of tRNAs anticodon stem loop cognate for Gly (residue 25–46) (ID4KR2), Arg (residue 25–46) (ID2ZUF) and Asp (residue 25–46) (ID1ASY) were used. The complete stem I domain of T-box riboswitch (4TZZ), which binds to RNA molecules, was used as a control. The following docking experiments were performed: tRNAs anticodon loop with the ancestral PTC; the complete stem I domain of T-box riboswitch (4TZZ) with the primitive PTC; the portion of the T-box riboswitch that binds to RNA with the primitive PTC; and the tRNAs anticodon loop with the modern PTC. The docking experiments were carried out with the support of Hex 8.0.0 software [21], according to the following parameter settings: correlation type—shape + electro + DAR S; FFT mode—five dimensions; sampling method—range angles; post processing—DARS minimization; grid dimension—0,6 solutions–2000; receptor range—180 step size–7.5; ligand range—180 step size–7.5; twist range—360 step size–5.5; distance range—40 box size–10; Translation step—0.8 sub step–0.0.

## 3. Results and Discussion

The ribosome emerged as a protagonist of the process of transition from an RNA world to a ribonucleoprotein world. In order to elucidate the origin and the primordial functions of the translation system, the 3D structure of ancestral PTC was reconstructed from ancestral sequences of tRNAs as suggested by Farias et al. [8]. Figure 1 shows the structural model obtained for the ancestor of the PTC, as well as the structural alignment of the model with the template obtained from *T. thermophilus*. The alignment between the ancestral PTC and the PTC of *T. thermophilus* involves 166 positions from which 87 positions showed exact base matches (52.4%). The structures were superimposed and showed a RMSD (root-mean-square deviation between P atoms) of 0.65 and a PSI (percentage of structural identity) of 0.927.

Note that the similarity of the sequences is 52.4% and the PSI has a high value of 0.927. Thus, the reconstruction of the 3D structure model by homology suggests that the tertiary structure of PTC has been highly conserved in evolution, presumably due to its central role in protein synthesis. These data are in line with the work of Petrov et al., who demonstrated that the PTC was conserved along the tree of life, and the process of ribosome evolution occurred by aggregation of parts around the catalytic center [1,2,7]. These results also reinforce the notion that the PTC could have originated from primitive tRNAs molecules, which worked together since the beginning of the earliest cells and orchestrated the emergence of the translation system [8,14,17]. From the data obtained by the homology modeling of the PTC, molecular docking experiments were performed with the ancestral PTC and tRNA^Gly^, tRNA^Arg^ and tRNA^Asp^ anticodon loops. The docking experiments results are shown in Figure 2.

The super-imposition of docking between the ancestral PTC and the three tRNAs analyzed is illustrated in Figure 3. As control, the docking experiment were done between the complete stem I domain of T-box riboswitch (4TZZ) with the ancestral PTC, as well as the portion of the T-box riboswitch that binds to RNA (position 36–71) with the ancestral PTC. In both cases, binding with the ancestral PTC was not observed.

These results suggest that the binding between the tRNA anticodon loop with the ancestral PTC is not random. Here, it can be observed that each of the chosen tRNAs interacted with different portions of the ancestral PTC, but in all cases the interactions never occurred with the anticodon, as it interacts with the PTC in modern organisms. This type of interaction made possible the approximation between tRNAs, where the primitive ribosome was the meet point and this affinity could have emerged by chemical affinity between the tRNAs and PTC. To test how this affinity was important in the emergence of the translation system, we performed docking experiments between the tRNA anticodon loop and the modern PTC (Appendix A). The following binding energies were obtained: tRNA^Gly^ (eTotal = −564.66 KJ/mol), tRNA^Arg^ (eTotal = −820.38 KJ/mol), and tRNA^Asp^ (eTotal = 2113.34 KJ/mol). The results show that the affinity between the tRNA anticodon loop and the PTC decreased in the evolutionary process. From this data, it can be suggested that the accretion of new parts in the primitive ribosome, as proposed by Petrov et al. [7], the evolutionary forces acted in the formation of peptides bond function and made possible the emergence of tRNA delivery system in the ribosome by elongation factors. In this manner, the binding energy between the tRNA anticodon loop and the ancestral PTC decreased the entropy in the system and increased the likelihood of peptides synthesis by the approximation of tRNAs charged with amino acids. Thus, the binding of the primitive acceptor arm to different parts of the PTC was possible and this entails the synthesis of small peptides without the need of a genetic code. The results suggest that in the emergence of the ribosome, the PTC must have acted as an adhesion center of for smaller RNA molecules, such as tRNAs charged with amino acids. These data agree with the proposal of Agmon et al. [9] and Belousoff et al. [10], who suggested that the primitive PTC worked as a simple ribozyme, randomly producing peptide chains. Herein, it is suggested that the amino acids were delivered by primitive tRNAs. The peptides synthesis without the need of a code, suggest that in the early stages of the process the anticodon had other functions: for example, they could be involved in the establishment of the assignments between amino acids and tRNAs. Thus, the emergence of the first genes or proto-genes must have reorganized the modes of interaction between tRNAs and PTC, which defined the sites of interaction A, P, and E in the modern ribosome. Hence, the anticodons were co-opted to stabilize the binding with the proto-gene and from this secondary interaction, the genetic correspondences between codons and amino acids were gradually established. In this work, we propose that the adhesion process of tRNAs in the ancestral PTC was an essential step in the origin of the genetic code, allowing that other RNAs with proto-gene functions, could interact with the anticodon loop. Based in the structure of the ancestral PTC, the gradual addition of structures as proposed by Petrov et al. must have been selected to increase the efficiency of the interactions between tRNA, PTC, and proto-genes or else to restrict the interaction of tRNAs to specific sites in the PTC [7]. Hence the emergence of the biological genetic code is an additional step in the formation of a primitive translation system, being preceded for peptides synthesis in a random fashion, already obeying an operational code. However, the delivery of amino acids in the primitive ribosome must have occurred via primitive tRNAs and not obtained directly from the system. Thus, it is necessary to re-examine the operational code model in this new context.

## 4. The Operational Code Revisited

Some authors suggested that the amino acid-accepting stem emerged before the anticodon loop of tRNAs, so that the first codification obeyed an operational code where amino acids were attached to their respective tRNA without the need of anticodon loop recognition [22,23,24,25]. Caetano-Anollés et al. have recently suggested that the aminoacyl-tRNA synthetases initially showed only the amino acid binding domain and that the domain of the anticodon of the tRNA recognition was a subsequent acquisition during the evolution of the system. In this context, the origin of the operational code is directly related to the absence of the anticodon loop in tRNAs, which enabled the first peptides to be synthesized in the absence of a genetic code [26]. However, Shimizu showed that small tRNAs with the portion of the anticodon loop could bind the amino acid as well [27]. Farias et al. suggested that the coding system was assembled by co-evolution between tRNAs and aminoacyl-tRNA synthetases, being driven by changes in the second base of the anticodon of tRNA, which in turn changed the hydropathy of the anticodon, and this pressure guided the diversification of aminoacyl-tRNA synthetases [28]. From this perspective, the anticodon loop was already present in primitive tRNA and should have been important in establishing specific interactions between tRNAs and their corresponding aminoacyl tRNA synthetases. The present results indicate the initial existence of an operational code due to the adhesion capacity of tRNAs without the presence of a genetic code in the PTC. This suggests that the anticodon loop initially increased the specificity between tRNAs and amino acids, and after the emergence of the proto-genes, by an exaptation process, the anticodon loops were co-opted to interact with the proto-genes, and thus the genetic code emerged and started to decode the biological information. From this point of view, this study proposes that the emergence of the operational code and the genetic code occurred simultaneously and that these systems played complementary roles in the origin and evolution of the translation system.

## 5. Conclusions

The results of this study indicate that the 3D structure of the PTC is highly conserved. The data also suggests that in the origin of the translation system the PTC worked as an adhesion center for tRNA molecules, which permitted the approximation of tRNAs charged with amino acids, thereby synthetizing peptides without the need of a genetic code. This also suggests that the organization of the sites A, P, and E of modern ribosomes occurred after the appearance of the first proto-genes, that organized the process of interaction between tRNAs and the PTC. Therefore, the emergence of the genetic code must have occurred as an exaptation process, where anticodons would initially have the function of increasing the specificity between amino acids and tRNAs, and that with the appearance of proto-genes would have been co-opted to establish a correlation between the information contained in the nucleic acids to proteins.

## Figures and Tables

**Figure 1 life-07-00021-f001:**
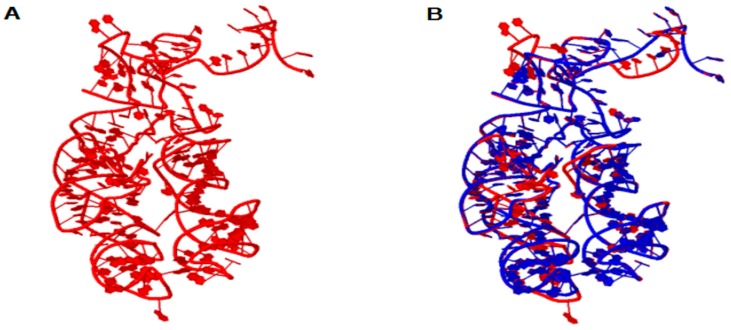
In (**A**) the 3D model of the ancestral Peptidyl Transferase Center (PTC). In (**B**) the structural alignment between the ancestral PTC (red) and the PTC of *T. thermophilus* (blue). The parameter values of the structural alignment are: RMSD = 0.65 and PSI = 0.927.

**Figure 2 life-07-00021-f002:**
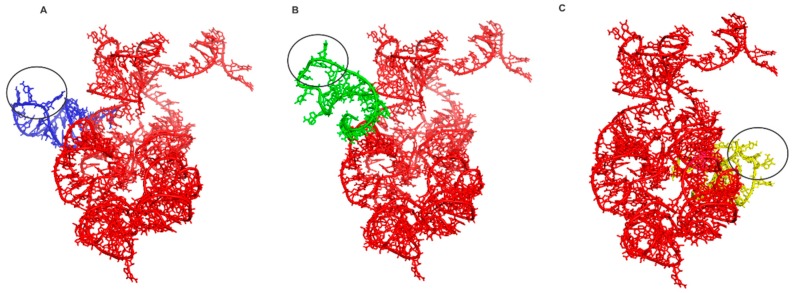
The docking experiments results. In red the ancestral PTC. In (**A**) the tRNA^Gly^ anticodon loop and PTC (eForce = −1929.96 KJ/mol), in (**B**) tRNA^Arg^ anticodon loop and PTC (eForce = −26017.11 KJ/mol) and in (**C**) tRNA^Asp^ anticodon loop and PTC (eForce = −5100.29 KJ/mol). The black circle indicates the anticodon.

**Figure 3 life-07-00021-f003:**
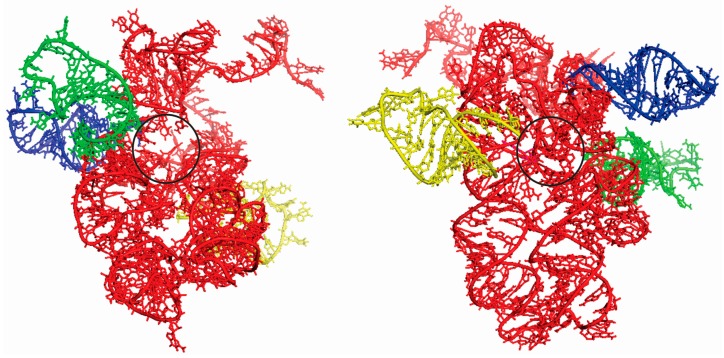
Two different perspectives of the superimposition of the docking experiments between the ancestral PTC (red) and tRNA^Gly^ (blue), tRNA^Arg^ (green) and tRNA^Asp^ (yellow) anticodon loop, with rotation of 180° between them. The black circle indicates the region of the exit tunnel to the peptides in formation.

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
