# Peer review of "Peptidyl Transferase Center and the Emergence of the Translation System"

_life, 2017, doi:10.3390/life7020021_

Round 1

Reviewer 1 Report

This paper is an ongoing follow up of an earlier FEBS Openbio paper.

In general the question of the origin of the PTC and how it developed over time, although largely ignored, is likely in fact one of the three most important questions in origins research. Thus, I can not fail to like this paper even if some details are not fully to my liking. The authors  envision a collection of tRNAs as having comed together originating what they now call the ancestral PTC.  This is mildly confusing as it might also be interpreted as stemming from the alignment of a large sample of 23S rRNAs and so this distinction should be made very clear.    .

In this paper, the authors explore this earlier ancestral PTC model further using well described molecular dynamics simulations. This is a logical next step.  They find somewhat surprisingly that different tRNAs dock in different places. They indicate that they did similar docking experiments with the modern PTC but I don't see the results of this study as to where the three different tRNAs bind. I think that if this data is available that it is an important addition.

Another consideration is the entrance to the exit tunnel (Fox et al. Astrobiology 12: 57-60 (2012)). This exit pore is essential to any ribosome as the growing peptide must have a path by which it leaves the PTC. In the context of the current paper, it is appropriate  to consider whether  the "ancestral" PTC exit hole (does it have one?) closely resembles that of the modern ribosome.  Also, where are the docked tRNAs in Figure 3 relative to the exit hole.   

Author Response

We thank the comments and suggestions made by reviewer

Comments and Suggestions for Authors

In general, the question of the origin of the PTC and how it developed over time, although largely ignored, is likely in fact one of the three most important questions in origins research.  Thus, I cannot fail to like this paper even if some details are not fully to my liking. The authors envision a collection of tRNAs as having come together originating what they now call the ancestral PTC. This is mildly confusing as it might also be interpreted as stemming from the alignment of a large sample of 23S rRNAs and so this distinction should be made very clear.

Answer

In order to build the ancestral PTC from tRNAs concatamers, we first reconstructed the ancestral sequences for each type of tRNA and made alignments to compare them with the modern PTC of T. thermophilus. The statistical analyses were made and described in: Farias ST, Rêgo TG, José MV. Origin and evolution of the Peptidyl Transferase Center from proto-tRNAs. FEBS Open Bio. 2014 Feb 8;4:175-8. doi:10.1016/j.fob.2014.01.010.

In this work, we determined the secondary and tertiary structure of the ancestral PTC and compare it with modern PTC (from T. thermophilus).

In this paper, the authors explore this earlier ancestral PTC model further using well described molecular dynamics simulations. This is a logical next step. They find somewhat surprisingly that different tRNAs dock in different places. They indicate that they did similar docking experiments with the modern PTC but I don't see the results of this study as to where the three different tRNAs bind. I think that if this data is available that it is an important addition.

Answer 

We added a figure in Supplementary Material showing the docking results between modern PTC and the tRNAs anticodon stem loops from Gly, Arg, and Asp.

Another consideration is the entrance to the exit tunnel (Fox et al. Astrobiology 12: 57-60 (2012)). This exit pore is essential to any ribosome as the growing peptide must have a path by which it leaves the PTC. In the context of the current paper, it is appropriate to consider whether the "ancestral" PTC exit hole (does it have one?) closely resembles that of the modern ribosome. Also, where are the docked tRNAs in Figure 3 relative to the exit hole.

Asnwer 

We indicated in Figure 3 (black circle), the position of the exit tunnel in the ancestral PTC.

Reviewer 2 Report

The authors have reconstructed the 3-D structure of the ancestral peptidyl transferase center based on the concatamers of ancestral tRNA sequecnes in order to test its interactions with the anticodon loop of tRNAs.  The authors then compared the 3-D structure of ancestral PTC and the current PTC in Thermus thermophilus.  The authors indicated that the PTC could have originated from primitive tRNA molecules that worked together since the beginning of the earliest cells and the translation system. The authors further concluded that the synthesis of small peptides did not need the genetic code.  I believe the authors have provided sufficient background information and explained the methods well. I only have three minor issues that I could like to suggest to the authors:

Line 28, missing a word "in"?

I understand that the reconstruction of the 3-D structure of PTC is based on the sequence available from another paper by the authors, please provide some details of how the 3-D structure was done.

The species name "Thermus thermophilus" needs to be italicized.

Author Response

We thank the comments and suggestions made by reviewer.

Comments and Suggestions for Authors

The authors have reconstructed the 3-D structure of the ancestral peptidyl transferase center based on the concatamers of ancestral tRNA sequences in order to test its interactions with the anticodon loop of tRNAs. The authors then compared the 3-D structure of ancestral PTC  and the current PTC in Thermus thermophilus. The authors indicated that the PTC could have originated from primitive tRNA molecules that worked together since the beginning of the earliest cells and the translation system. The authors further concluded that the synthesis of small peptides did not need the genetic code. I believe the authors have provided sufficient background information and explained the methods well. I only have three minor issues that

I could like to suggest to the authors:

Line 28, missing a word "in"?

 Answer

We added “in”

I understand that the reconstruction of the 3-D structure of PTC is based on the sequence available from another paper by the authors, please provide some details of how the 3-D structure was done.

 Answer

In Materials and Methods, we describe how we obtained the 3D-structure. Briefly, the reconstruction of the ancestral structure uses an alignment between the ancestral sequence and the modern sequence. The modern sequence of PTC is also provided. From this data, the program calculates the secondary structure and the tertiary structure by homology. The Parameter default values were used.

The species name "Thermus thermophilus" is now in italics.